# Coconut Callus Initiation for Cell Suspension Culture

**DOI:** 10.3390/plants12040968

**Published:** 2023-02-20

**Authors:** Eveline Y. Y. Kong, Julianne Biddle, Sundaravelpandian Kalaipandian, Steve W. Adkins

**Affiliations:** 1School of Agriculture and Food Sciences, The University of Queensland, Gatton, QLD 4343, Australia; 2The Queensland Alliance for Agriculture and Food Innovation (QAAFI), Centre for Horticultural Science, The University of Queensland, Indooroopilly, QLD 4068, Australia; 3Australian Centre for International Agricultural Research, Canberra, ACT 2617, Australia

**Keywords:** callus, cell suspension culture, coconut, *Cocos nucifera* L., embryogenic callus, tissue culture, somatic embryogenesis

## Abstract

The development of a cell suspension culture system for the scaling up of coconut embryogenic callus (EC) production would drastically improve efforts to achieve the large-scale production of high-quality clonal plantlets. To date, the hard nature of coconut EC appeared to be the main constraint for developing cell suspension cultures. Hence, this study attempted to acquire friable EC through the following approaches: The manipulation of (1) medium type and subculture frequency, (2) a reduced 2,4-dichlorophenoxy acetic acid concentration during subculture, (3) the nitrate level and the ammonium-to-nitrate ratio, and the addition of amino acid mixture, (4) the addition of L-proline, and (5) the reduction of medium nutrients. Unfortunately, none of these culture conditions produced friable coconut EC. Even though friable EC was not achieved via these approaches, some of the conditions were found to influence the formation of compact EC, therefore these results are important for further studies focused on somatic embryogenesis in coconut and other species.

## 1. Introduction

Coconut (*Cocos nucifera* L.) is a recalcitrant species and difficult to work with in in vitro culture, particularly to undertake plant regeneration via somatic embryogenesis (SE) [1]. However, given the growing demand for the rapid production of high-quality coconut planting materials with characteristics such as early fruit bearing, high yield, and disease resistance, micropropagation via SE is now considered to be a way to rapidly produce ‘cloned’ coconut plantlets on a large scale [2]. Nevertheless, significant improvements to the existing SE protocol for coconut are still necessary before plantlets can be produced on a large scale. The steps in coconut SE currently include the production of embryogenic callus (EC), the formation of somatic embryos, and their germination [3].

The introduction of a liquid culture system, that is, a cell suspension culture step, to scale up the production of somatic EC could be an important improvement to the protocol. In general terms, the benefits of using a liquid culture medium have been shown to outweigh those of using a semi-solid culture medium [4]. Previous studies have reported 5400 and 10,000 somatic embryos can be produced from each litre of the cell suspension culture medium for oil palm (*Elaeis guineensis* Jacq.) [5] and date palm (*Phoenix dactylifera* L.) [6], respectively. In a study on date palms, the somatic embryo production from SE suspension cultures within 2 months was ca. 5-fold greater than for the yield from a semi-solid medium after 4 months [7]. Hence, a somatic embryogenic cell suspension culture step is often extremely important to achieve rapid cell production with sustained embryogenic potential and to ensure synchronized somatic embryo production [8].

To date, the typical kind of callus produced from coconut explants is hard in nature, and this characteristic is the likely reason it has been difficult to initiate high-quality coconut cell suspension cultures [9]. Hence, no high-quality coconut cell suspensions have yet been successfully established, and there is only one instance known where friable callus has been reported [10]. This presents a challenge because callus friability appears to be a prerequisite for the establishment of an embryogenic cell suspension culture for coconut as it is for many other species, including similar species such as oil palm and date palm [8]. In general, friable callus is considered to be the preferred inoculum for the initiation of cell suspension cultures in most species, as this kind of callus falls apart more easily when transferred to a liquid medium where agitation helps to disintegrate the cell aggregates [11]. In addition, compact callus generally has a poor capacity to proliferate as it is developing the ability to differentiate [12].

The focus of this study is to determine which factors can encourage the formation of friable EC. Experiments were designed based on previous studies undertaken on coconut and other species [red goosefoot (*Chenopodium rubrum* L.), date palm, sorghum (*Sorghum bicolor* L. Moench), maize (*Zea mays* L.), and cassava (*Manihot esculenta* Crantz)]. The factors that had been shown to affect friable callus or friable EC formation in these studies were the medium form (liquid or semi-solid) [13], the frequency of subculture [10,14], the concentration of 2,4-dichlorophenoxyacetic acid (2,4-D) during subculture [15,16], the nitrogen level and form [12], the addition of an amino acid mixture [17] or L-proline [18,19], and the use of reduced nutrient content in the medium [20]. Hence, these factors were considered in the present study with the aim to produce friable EC in coconut. Plumule tissue produced from germinating coconut zygotic embryos and compact EC produced from plumules were the starting explant materials used, and histological analysis was conducted to determine the anatomical differences in the callus types formed.

## 2. Results

### 2.1. Medium Type and Subculture Frequency for the Formation of Friable and Compact EC (Experiment 1)

The first experiment was designed to determine the influence of the medium type (liquid or semi-solid) and subculture frequency (subculture every 2 or 4 weeks or no subculture for 20 weeks) on the formation of friable EC. These were trialled because a liquid culture medium has been demonstrated to promote better growth than a semi-solid medium as the plants can absorb water and nutrients more rapidly [4]. Callus subculture frequency is another important factor, as a higher proportion of coconut EC has been shown to form when cultures went undisturbed for a period of 6 months [14].

The results showed that the medium type and subculture frequency were not critical factors for the induction of friable EC, as it was not formed in either treatment (T). The control, which followed the protocol of Nguyen [10] but used primary callus instead of secondary callus, did not produce friable EC. According to this earlier protocol, friable EC would form from secondary compact EC after 5 months of culture. Instead of the friable EC being formed in this experiment, compact callus of either an embryogenic (Figure 1A) or non-embryogenic (Figure 1B) nature were formed on the Eeuwens Y3 (Y3) [21] medium from the primary EC.

Embryogenic callus was identified by its smooth, translucent, and well-defined somatic structure, while the non-EC was found to be rough, yellowish, and sponge-like. The compact callus clumps formed from the liquid culture were black in colour due to a covering of activated charcoal (AC; Figure 1C). The callus clumps formed within liquid media were all found to be non-embryogenic, made up of rough callus, with no distinct or well-defined structures.

Interestingly, the formation of compact callus and EC was affected by the medium type and subculture frequency (Figure 2). The formation of callus (determined by the number of explants forming callus from the total number cultured) was significantly increased (40.0 ± 8.5%) on a semi-solid medium (T2), but it was significantly reduced (3.3 ± 3.3%) when cultured in a liquid medium (T3). A higher subculture frequency (T4; 33.3 ± 8.2%) produced more callus from explants inoculated in a liquid medium than those not sub-cultured (T3).

The liquid medium significantly reduced the formation of compact EC regardless of the subculture frequency (Figure 2). Hence, the semi-solid medium was found to be more suitable to produce compact EC than the liquid medium. As for the compact EC formation on the semi-solid medium between sub-cultured (control and T1) and no subculture (T2) treatments, the production of EC from T2 was 16.7 ± 5.3%, whereas the control and T1 had only 3.3 ± 3.3% EC formed (Figure 2). Hence, the result suggested that no subculture of explants on semi-solid medium was better for the formation of compact EC.

### 2.2. 2,4-D Concentration for the Formation of Friable and Compact EC (Experiment 2)

As the production of EC is primarily due to the addition of auxin, the formation of friable callus may be closely linked with the concentration of 2,4-D used. For instance, a reduction in the 2,4-D concentration in the subculture medium in previous experiments has been found to increase the friability of the coconut EC derived from both seedlings and young inflorescence tissues and resulted in the formation of friable and well-defined EC from nodular callus [15]. Furthermore, a low 2,4-D concentration (0.1 mg L^−1^) was found to induce friable EC from juvenile date palm leaves [16].

Hence, the reduction of 2,4-D concentration was tested in this experiment, from 600 to 6 µM in the subculture medium, with the intention to induce the formation of friable EC from compact EC (explant). However, after two subculture cycles, each of 2 months, there was no friable EC observed in either of the 2,4-D treatments (6 or 600 µM). After the 2,4-D concentration was reduced to 6 µM in the subculture medium (Table 1), mature somatic embryos (58%) and roots (20%) were developed (Figure 3) from the compact EC at a significantly higher rate when compared with the control (600 µM 2,4-D) as shown in Table 1. Non-EC was also formed (98%), but none of the explants remained as compact EC after being sub-cultured in media containing 6 µM 2,4-D (Table 1). Moreover, the percentage of explants that necrosed after subculture was significantly lower for this treatment (3%; Table 1). In the control, the explants either necrosed or formed callus (embryogenic or non-embryogenic; Table 1). However, most of the explants formed were non-EC (80%) and only 32.5% remained as compact EC (Table 1). Mature somatic embryos or roots were not observed in the control. At the end of this experiment, the explants with mature somatic embryos and roots were continuously sub-cultured every 4 weeks in a plant growth regulator (PGR)-free Y3 medium. The mature somatic embryos and roots continued to grow and elongate and eventually formed plantlets (Figure 4). This result indicated that a reduction of the 2,4-D concentration during subculture can lead to the formation of mature somatic embryos (58%) and eventually plantlets without the use of cytokinin. This information is useful for developing an optimized somatic embryo maturation medium for coconut SE.

### 2.3. Change in Nitrogen Presentation for the Formation of Friable and Compact EC (Experiment 3)

The type of EC formed in sorghum (*Sorghum bicolor* L. Moench) is known to be affected by the concentration of ammonium ions in the medium [12]. In a study undertaken on sorghum callus [22], friable EC was formed in a medium containing a low concentration of nitrate along with a high nitrate-to-ammonium-ion ratio (4:1), and both friable and compact EC were induced in a medium containing a high concentration of nitrate when a nitrate-to-ammonium-ion ratio of 1.9:1 was used. This is likely to be applicable to coconut because the concentration of inorganic nitrogen is one of the most important factors affecting coconut tissue growth, as coconut tissues have a specific requirement for ammonium ions and a high concentration of nitrate [21]. In addition, the initiation of SE in coconut is linked to high uptake of ammonium, calcium, magnesium, and sucrose from the medium [23], and amino acids are involved in the metabolism of nitrate and ammonium ions [24]. Hence, the addition of a mixture of amino acids was studied as it might be able to be used to supplement the high nitrogen needs of coconut tissues.

In the Y3 medium formulation, potassium nitrate and ammonium chloride are the sources of nitrogen. The control used was the usual Y3 medium, with no concentration change in the nitrogen level. Based on the results, increased concentrations in these nitrogen compounds, the addition of an amino acid mixture (arginine, glutamine, and asparagine), and the higher ratio of nitrate to ammonium ions presented in the Y3 medium (Section 4.6) did not result in the formation of friable EC.

Instead, compact callus of either an embryogenic or non-embryogenic nature was formed from all treatments (Figure 5). The control, T1, T2, T3, and T5 were not significantly different from each other in terms of the percentage of callus formed (ranging from 98.3 to 100%), whilst an increase in the concentration of nitrate and ammonium with a ratio of 2:1, without the addition of an amino acid mixture (T4), produced a significantly lower percentage of callus (26.7%) when compared to all other treatments (control, T1, T2, T3, and T5; Figure 6).

Regarding the percentage of compact EC formed (Figure 6), the treatments were not significantly different; however, T1, T3, and T5, which were supplemented with the amino acid mixture, showed a higher percentage than those without this addition (given the concentration of nitrate and ammonium ions and its ratio were the same). For example, the percentage of compact EC of T1 (30.0%) was higher than the control (25.0%), T3 (13.3%) was higher than T2 (8.3%), and T5 (18.3%) was higher than T4 (5.0%). Hence, the addition of amino acids had a positive effect on EC formation. However, an increase in the nitrogen ions concentration and the ratio of nitrate to ammonium ions (T2), as well as the supplementation of nitrate and ammonium ions at a higher concentration without a change in its ratio (T4), resulted in a lower percentage of compact EC being formed. Although friable EC was not induced with the change in nitrogen level, the results indicated that an increased nitrate and ammonium concentration up to a ratio of 4:1 (T4) was not beneficial for the production of compact EC, and T1 supplemented with an amino acid mixture was the most successful treatment for the induction of compact EC.

### 2.4. L-Proline and Medium Type for the Formation of Friable and Compact EC (Experiment 4)

The addition of L-proline can also affect friable EC formation in the case of maize (*Zea mays* L.). Friable maize EC was found to form in an N6 [25] medium (with a nitrate-to-ammonium-ion ratio of 4:1) supplemented with L-proline, while a Murashige and Skoog (MS) [26] medium (with a nitrate-to-ammonium-ion ratio of 1.9:1) supplemented with or without L-proline could not induce friable EC [18]. Although the exact role of L-proline is not clear, it was found to be essential for the initiation of friable EC in maize [18]. This was confirmed by another study using maize tassel segments as explants. The addition of L-proline was critical for the formation of friable EC, regardless of the medium used (MS or N6 medium) [19]. This could also be the case for coconut, as Eeuwens [21] mentioned that the inorganic nitrogen source and concentration were perhaps the most critical factors for the difference in coconut growth in vitro. This is because it plays an important role in plant growth, tissue differentiation, and development, including the stimulation of storage protein synthesis. Most importantly, it is a determining factor in cell wall protein production, which is important for cell wall signal transduction, plant development, and stress tolerance [27].

Hence, an experiment was conducted to study the effect of L-proline on the formation of friable EC. Based on the results, the addition of L-proline into different treatments was unable to induce the friable EC from coconut plumules. Instead, compact EC and non-EC were formed after 12 weeks of culture. The compact EC formed from plumule tissues of the CAIRNS variety was shown in Table 2. In general, the formation of compact callus from all treatments was high as the percentage ranged from 90.0 ± 5.8 to 100%. On the other hand, the mean percentage of explants producing compact EC was considerably lower. The control, which consisted of the Y3 medium and 600 µM of 2,4-D without the addition of L-proline, recorded the highest compact EC formation percentage (13.3 ± 6.7%), followed by the compact EC formation percentage of T4 (6.7 ± 3.3%), but these values were not significantly different to the percentages of T2 and T3, in which no compact EC was formed. Overall, L-proline had a negative effect on the formation of compact EC, and it had no influence on the induction of compact non-EC callus, but the exchange of ammonium chloride with ammonium sulphate in the Y3 medium supplemented with L-proline led to the production of compact EC at a low percentage.

### 2.5. Effect of Reduced Nutrients on the Formation of Friable and Compact EC (Experiment 5)

The lack of friable EC production for coconut could be due to the concentration of mineral nutrients used for coconut culture. In the study by Taylor et al., in 1996 [20], it was found that changes in the concentration of picloram, casein hydrolysate, L-proline, and glutamine, as well as sucrose, did not lead to the formation of friable EC from the embryogenic structures of cassava (*Manihot esculenta* Crantz). However, friable EC was successfully induced from embryogenic structures cultured on a Gresshoff and Doy (GD) [28] medium. This medium has a lower macro- and micro-nutrient content compared to MS or Eeuwens Y3 [21] media. Structural changes in cell walls are thought to be the main factor that aids in the transition from somatic to friable EC because they loosen cell wall attachments in the somatic embryo and allow for the induction of friable EC [29].

Hence, the effect of reduced nutrients on the formation of friable EC was considered in this study. Based on the results, there was no friable EC produced from the primary compact EC, which was used as the explant for this experiment. The induction of friable EC was not initiated by the reduction of nutrients in the media. Instead, secondary compact callus with embryogenic and non-embryogenic potential was produced for both Indian coconut varieties tested (Indian Yellow Dwarf (IYD) and Sampoorna; Figure 7). The compact EC formed for both varieties was similar in appearance (Figure 8 and Figure 9) but differed in their response to similar treatments, that is, in terms of the percentage of explants with secondary compact EC and browning of tissues. Necrosis or browning also occurred in the primary compact EC, which was used as the explant to further induce secondary callus (Figure 8 and Figure 9). This necrotic response was also observed in the treatments for the Sampoorna variety (Figure 7).

For the IYD variety, T2 produced significantly more secondary compact EC (63.3 ± 6.7%) compared to T1 (36.7 ± 3.3%; Figure 7). This indicated that there are more explants (primary compact EC) that became non-EC after culture in the Y3 media supplemented with only 10% of N, P, and K for 8 weeks in T1. The necrotic response for secondary compact EC for this variety was not significantly different between T1 and T2 (37 to 63%).

In contrast, the Sampoorna variety had a different response to the treatments in terms of the percentage of explants with secondary compact EC as well as the percentage of explants with browning of tissues (Figure 7). All the explants in the control produced secondary compact EC after 8 weeks of culture (Figure 7), whilst T1 (83.3%) and T2 (63.3%) had a negative effect on the production of secondary compact EC, with the values significantly lower than the control (100%). In contrast to the IYD variety, T1 had a significantly higher percentage of explants with secondary compact EC than T2 for the Sampoorna variety. This suggested that T1 had fewer explants (primary compact EC) transformed into non-EC after the treatment. Regarding browning, it was more significant for explants in the control (63.3 ± 12.0%) and T1 (56.7 ± 3.3%) than for explants in T2 (23.3 ± 3.3%). The results on both coconut varieties suggested that reduced nutrients were not suitable for the growth of compact EC, and its effect on compact EC formation was dependent on the variety used. It was also confirmed that the Y3 medium was a better medium than the GD medium for the formation of compact EC.

### 2.6. Histological Analysis

Histological analysis was used to determine the anatomical differences in the callus types formed. Compact EC and non-EC were found to be easily differentiated with histological analysis. Compact EC (Figure 10) was rich in embryogenic cells, which were actively dividing. Embryogenic cells were characterised by the presence of large nuclei and easily visible nucleoli in the cytoplasm, which were surrounded by many starch granules. Starch granules and cell walls were stained pink by the periodic acid Schiff’s reagent, while the nucleus and nucleolus were stained blue. In contrast, non-EC (Figure 11) did not have meristematic cells, and the cells in the central area appeared to be parenchymatic. Most of the cells were smaller in size when compared to embryogenic cells. The non-embryogenic cells were characterised by the presence of small nuclei, which were stained blue, while their nucleoli were not visible and starch granules were not present.

## 3. Discussion

The creation of a cell suspension culture step in coconut micropropagation is likely to be dependent on the formation of friable EC. As a result, studies were undertaken on the influence of several treatments that may stimulate the formation of friable EC. These approaches included varying the medium type [13], the subculture frequency [10,14], reducing the medium 2,4-D concentration [15,16], manipulating the nitrate level, changing the ammonium-to-nitrate ratio [12], supplying L-proline [18,19], and reducing the medium nutrient level [20], all approaches that had worked for friable EC production in other species. However, none of these approaches was able to stimulate the formation of friable EC for coconut.

The possible reason for the lack of formation of friable EC for coconut with these treatments may be due to the initially very hard nature of the coconut EC, as has been previously suggested by Bhavyshree et al., 2016 [9], or it may be due to the explant type that was used. To date, very few studies [9,10,30] have attempted to initiate cell suspension cultures from coconut callus, and only one [10] has reported being successful in producing friable EC, but it has not been successfully repeated.

It is possible that the formation of friable EC is linked to a dedifferentiation process, hence, the reason for success in the one study may have been because a 5-month-old secondary EC was used [10], while primary EC was mainly used in the experiments reported in this study. Dedifferentiation most often occurs in aged or damaged tissues. Therefore, it is important in the future to examine additional factors that can promote dedifferentiation, such as culture stress and age [31]. In addition, it will be important to determine the influence of using a different explant type, such as somatic embryos, to induce friable EC in coconut. The use of cassava (*Manihot esculenta* Crantz) somatic embryos to produce friable EC was associated with a dynamic change in cell structure, physiology, and gene expression [29]. Hence, cell reconstruction by loosening the cell walls is likely to be an important event if friable EC is to be induced from coconut somatic embryos in future research.

Although there was no friable EC formed in any experiments in this study, some of the factors tested were found to stimulate the formation of compact EC, therefore this study is useful for further work on SE. Firstly, a semi-solid medium was found to be better than a liquid medium for inducing compact EC from coconut plumule tissues, and a long subculture cycle was preferential for achieving this (Figure 2). This was possibly due to the presence of the gelling agent in the semi-solid medium, as the agent increased the medium pH and lowered the adsorption of 2,4-D by AC [32,33,34]. As a result, the 2,4-D availability was higher in the semi-solid medium than in the liquid medium, thereby improving the formation of compact EC. Previous work had already shown that in the presence of AC, a high level of 2,4-D (600 µM) is required to stimulate compact EC formation (Table 1), while a low 2,4-D concentration (6 µM) was required for the formation, maturation (ca. 58%) and germination of somatic embryos to form shoots, then plantlets (Figure 4). Hence, it was found that the removal of 2,4-D from the medium triggered the maturation of somatic embryos. These observations became particularly useful for the study on the improvement of somatic embryo maturation. Apart from that, the addition of amino acids was found to positively affect the formation of compact EC from plumules (Figure 6), likely due to its function as a source of nitrogen, which generally improves cell growth rates while stimulating the morphogenic and embryogenic processes in plants [24]. In contrast, the addition of L-proline was found to negatively influence primary compact EC formation (Table 2). A reduction in nutrients was detrimental as it led to a loss of somatic embryogenic potential within the explants and resulted in callus necrosis and, ultimately, to its death (Figure 7). Results from these studies will be useful to help improve the formation rate of compact EC of all coconut varieties, especially those that have naturally low compact EC formation rates.

Histological studies of the compact EC and non-EC revealed that each callus type could be clearly identified by the presence or absence of starch granules, with starch granules being present in compact EC and absent in non-EC (Figure 10 and Figure 11). Hence, this created a tool that could be used for the swift identification of embryogenic cell lines, a tool that will be useful in future work examining embryogenic capacity in cell lines cultured in a bioreactor for lengthy periods of time. This is particularly important in a commercial setting, where bioreactors will be the main vessels to maintain the large volume of cells required for plantlet formation.

## 4. Materials and Methods

### 4.1. Plant Materials

The mature coconut embryos (ca. 12-month-old), used in all experiments, came from Xiem Xanh Dwarf (XXD) and Dua Dua (AR; an aromatic variety), both from Vietnam; Malayan Yellow Dwarf (MYD) from the Philippines; the fruit of unknown variety sourced from Northern Queensland, Australia (referred as ‘CAIRNS’); or Indian Yellow Dwarf (IYD) or Sampoorna (a hybrid cross between IYD and Andaman Tall) obtained from Deejay Coconut Farm in Karnataka, India. The embryos of XXD, MYD, AR, and CAIRNS varieties were used in the experiments reported in Section 2.1, Section 2.2, Section 2.3, and Section 2.4, respectively, while IYD and Sampoorna varieties were used in the experiment reported in Section 2.5. The variety used in each experiment was selected based on availability at the time of experimental planning.

Prior to transport to Australia, the Vietnam coconut varieties were collected from Ben Tre Province. The endosperm plugs, each containing a single embryo, were then extracted from the dehusked fruit and surface sterilized with 70% (*v*/*v*) ethanol for 3 min. Then, in a laminar air flow (LAF) hood, the embryos were isolated and surface-sterilized with 70% (*v*/*v*) ethanol for 1 min and further immersed in a 0.5% (*v*/*v*) sodium hypochlorite (NaOCl) solution for 5 min. The embryos were then rinsed three times with sterile distilled water and cultured individually onto a semi-solid PGR- and sucrose-free Y3 medium, supplemented with agar (6 g L^−1^) and AC (2.5 g L^−1^) contained within 2 mL centrifuge tubes. In the case of the shipment of AR embryos, no AC was added to the medium.

As for those from the Philippines, the zygotic embryos were provided by the Philippine Coconut Authority-Zamboanga Research Centre. The endosperm plugs were extracted using a fabricated drilling machine and washed under running tap water, then in a diluted liquid soap and rinsed again in tap water. The plugs were then surface sterilized with a bleach (which contained NaOCl) solution for 30 min followed by abundant rinsing in sterile water. Then, the zygotic embryos were isolated in a LAF hood and surface-sterilized with bleach for 5 min, followed by five-time rinsing with sterile distilled water. These embryos were cultured singly onto a semi-solid PGR-, sucrose-, and AC-free Y3 medium supplemented with agar (7.8 g L^−1^) contained within 2 mL centrifuge tubes.

The preparation of IYD embryos obtained from India followed a similar protocol to those obtained from Vietnam. However, upon isolation of Sampoorna embryos, they were surface sterilized with a 0.5% (*v*/*v*) NaOCl solution for 10 min rather than for 5 min. All embryos from India were individually cultured onto a semi-solid (agar 6 g L^−1^), PGR-free Y3 medium supplemented with sucrose (30 g L^−1^) and AC (2.5 g L^−1^) contained in 2 mL centrifuge tubes. These embryos were prepared in accordance with established protocols for transport but adapted depending on the availability of reagents and the preference of the staff preparing samples.

All overseas materials (from Vietnam, the Philippines, and India) were transported by airfreight to the approved quarantine facility [AQF; Plant Industries Building Room 101, Gatton Campus, The University of Queensland (UQ)] under the quarantine import permit.

### 4.2. Surface Sterilisation

Following transport from their source location, the embryos received underwent different sterilization steps based on the in-country preparation protocols that had been applied prior to shipping, the shipment duration, and the visual assessment of embryo quality upon receipt. Following clearance through Australian quarantine, the embryos from Vietnam and India (XXD and IYD) were removed from their vessels, rinsed three times with sterile distilled water, and used for experiments in Section 2.1 and Section 2.5, respectively. Once the embryos from the Philippines and Vietnam (MYD and AR) were cleared through quarantine, they were then removed from their vessels and re-surface sterilized in a 95% (*v*/*v*) ethanol solution for 3 min, followed by shaking for 20 min in a 0.5% (*v*/*v*) NaOCl solution prepared from fresh commercial bleach (which contained 42 g L^−1^ NaOCl). The embryos were then rinsed three times, each for 3 min in sterile distilled water. These embryos were used in Experiments Section 2.2 and Section 2.3.

Similar steps were performed on the Sampoorna embryos (experiment Section 2.5), but the re-surface sterilizing agents used this time were 70% (*v*/*v*) ethanol (1 min) a and 6% (*w*/*v*) calcium hypochlorite [Ca(OCl)_2_] solution (prepared from commercial pool chlorine consisting of 650 g kg^−1^ of Ca(OCl)_2_), for 5 min before triple rinsing in sterile distilled water.

For CAIRNS domestic coconuts, dehusked fruits were delivered by road transport from northern Queensland and used in experiment Section 2.4. Each fruit was split in half using a stainless-steel knife, and the endosperm plug containing the embryo (located under the ‘soft’ eye) was extracted using a clean 2 cm diameter cork borer. The endosperm plugs were first surface sterilized with 70% (*v*/*v*) ethanol for 3 min and then the embryos were carefully isolated in a LAF hood. The isolated embryos were then surface sterilized by shaking in 70% (*v*/*v*) ethanol for 1 min, followed by gentle swirling for 15 min in freshly prepared 2.1% (*v*/*v*) NaOCl. The embryos were then rinsed three times with sterile distilled water in the LAF hood.

After these various surface sterilization treatments, embryos from all varieties were inoculated individually into 30 mL polycarbonate tubes containing ca. 8 mL of a semi-solid (Gelzan 3 g L^−1^; Sigma, G1910) embryo germination medium (EGM), which consisted of Eeuwens Y3 nutrients [21], AC (Sigma, C9157; 2.5 g L^−1^), sucrose (30 g L^−1^), and 6-benzylaminopurine (BAP; Sigma, B3408; 5 µM). The embryos were incubated in the dark for 2 weeks at a constant temperature of 27 ± 1 °C. After this time, and from the healthy, germinated, and clean embryos only, plumules were isolated. To do this, each embryo was placed under a clean dissecting microscope in the LAF hood and cut across the upper surface, directly over the emerging plumule, into two equal halves. Then, the two halves of the plumule were excised from each embryo half and the cut surface of those plumule halves was placed down onto the surface of the culture medium as described in each experiment. A video of plumule isolation is available in the Appendix A.

### 4.3. Culture Medium and Plant Growth Regulators

Eeuwens Y3 (Y3) [21], Chu (N6) [25], and Gresshoff and Doy (GD) [28] media were supplemented with sucrose (30 g L^−1^), Gelzan (3 g L^−1^), and AC (2.5 g L^−1^) unless stated otherwise. The media were adjusted to pH 5.8 via drop-wise addition of either 1 M NaOH or HCl prior to autoclaving. All media and apparatus, including vessels, were sterilized in an autoclave for 20 min at 121 °C with a pressure of 2.0 kg cm^−2^.

The stock solutions of the PGRs were prepared mostly from powder, firstly by being dissolved in drops of 1 M sodium hydroxide then being topped up to volume using deionised water. Amino acid stock solutions were readily dissolved in distilled water in the preparation of stock solutions. Prior to addition to the autoclaved Y3 medium, all stock solutions were filtered and sterilized using a 0.22 µM filter membrane. The PGRs were added to the autoclaved media when it had cooled to ca. 50 to 55 °C.

### 4.4. Medium Type and Subculture Frequency for the Formation of Friable and Compact EC (Experiment 1)

After two weeks of culture, the plumules from the germinating embryos were isolated and cultured onto a callus induction medium (CIM) of either a liquid or a semi-solid form and kept under dark conditions at 27 ± 1 °C. The semi-solid CIM (36 mL) consisted of a Y3 medium supplemented with 2,4-D (Sigma, D7660; 600 µM), AC (2.5 g L^−1^), sucrose (30 g L^−1^), and Gelzan (3 g L^−1^) in 250 mL small sterile glass bottles. The liquid CIM was prepared as described above for the semi-solid CIM, but it did not contain Gelzan. The liquid medium (25 mL) was placed into sterile 100 mL wide-neck conical flasks.

For callus induction using liquid media, the media were kept unshaken for 14 days after culturing, then on day 15, 100 mL conical flasks were transferred to an offset orbital shaker operating at a speed of 80 rpm and increased to 120 rpm after 2 weeks. The cultures were maintained at 120 rpm for a further 16 weeks. This protocol was adapted from Geile and Wagner, 1980, [13] with a slight modification. The subculture process involved the removal of 15 mL of the old liquid medium using a pipette and the addition of fresh liquid CIM to bring the volume back up to 25 mL per conical flask. Those plumules with formed callus were transferred to fresh semi-solid or liquid CIM during subculture. The subculture interval for each treatment is shown in Table 3. The culture period for all treatments was 20 weeks, excluding the control, which had a 32-week culture period. The control adapted from Nguyen, 2018 [10], was used as it was considered the ‘best’ method developed to date for friable EC production. Although, instead of using secondary callus as in the study by Nguyen, 2018 [10], primary callus was used to induce friable EC due to the availability of starting materials for this study. The control in this study had a semi-solid medium approach, and the first step involved the culturing of the isolated plumule on semi-solid CIM to induce primary callus. The plumule with intact primary callus was then not sub-cultured for 12 weeks. Then, the single EC was separated from the primary callus and cultured back onto a fresh semi-solid CIM for another 20 weeks. The percentages of callus and compact or friable EC were recorded at the end of the culture period (20 weeks).

There were five replicates for each treatment, and each treatment consisted of a pair of culture vessels, which contained three explants (a half plumule represented one explant) per vessel. A randomized complete block design (RCBD) was used with the experiment blocked by time as the explants were isolated at different times. A two-way analysis of variance (ANOVA) was used for data analysis utilizing Minitab 17 Statistical Software.

### 4.5. 2,4-D Concentration for the Formation of Friable and Compact EC (Experiment 2)

Each half of the plumule was excised from an embryo and grown in semi-solid CIM containing the Y3 medium supplemented with 2,4-D (600 µM) for 3 months. The Petri dishes (90 × 25 mm diameter/height; 42 mL media in each Petri dish) containing the plant material were kept in a growth chamber in dark conditions with a temperature of 27 ± 1 °C. The compact EC were selected and used as explants in this experiment. Each explant (50 mg compact EC) was sub-cultured twice onto a fresh medium of the same constituents, and each cycle was 8 weeks in duration. The subculture medium was supplemented with one of two 2,4-D concentrations, 6 (treatment) or 600 µM (control). The control consisted of 600 µM 2,4-D. The cultures were observed after 28 weeks for the formation of friable EC. The shoots produced were then maintained on PGR-free Y3 medium under illuminated (40 µmol m^−2^ s^−1^) conditions with a photoperiod of 16/8-h (light/dark) at a temperature of 27 ± 1 °C and sub-cultured onto fresh medium every 4 weeks.

Each treatment had four replicates, and each replicate consisted of a pair of Petri dishes with five explants per Petri dish. The data collected for each parameter were analysed using a one-way ANOVA.

### 4.6. Change in Nitrogen Presentation for the Formation of Friable and Compact EC (Experiment 3)

The isolated plumule halves were cultured onto a Y3 medium containing different combinations of nitrate and ammonium ions, at different concentrations, with or without amino acids (Table 4). The combinations were selected based on a comparison of the concentration of nitrate and ammonium ions, their ratios, and total nitrogen based on previous studies on sorghum [22] and coconut [21]. The Y3 medium for each treatment was supplemented with 2,4-D (600 µM), AC (2.5 g L^−1^), sucrose (30 g L^−1^), and Gelzan (3 g L^−1^). The mixture of amino acids consisted of L-glutamine (Sigma, G8540; 685 µM), L-arginine (Sigma, A8094; 574 µM), and L-asparagine (Sigma, A4159; 667 µM) and the concentrations used were based on the formulations of Eeuwens [17]. Each amino acid solution was filter-sterilized and added to the medium after autoclaving. The culture conditions used were total darkness at 27 ± 1 °C, and plumule halves were incubated for 12 weeks in Petri dishes. Data on the percentage of callus formed and friable or compact callus formed were recorded after 12 weeks.

Each treatment consisted of five replicates with each replicate being made up of two Petri dishes, each with six explants per Petri dish and each explant consisted of two halves of the plumule. The experimental design was a RCBD with the experiment being blocked by time. A two-way ANOVA was used to analyse the data.

### 4.7. L-Proline and Medium Type for the Formation of Friable and Compact EC (Experiment 4)

After 14 days of culture on EGM, the embryos were cut in half and the plumules were excised from each embryo half. The isolated materials were cultivated in different CIM as shown in Table 5 and incubated under total darkness at 27 ± 1 °C. Each treatment consisted of media supplemented with 2,4-D (600 µM), AC (2.5 g L^−1^), sucrose (30 g L^−1^), and Gelzan (3 g L^−1^) and contained within Petri dishes. The filter-sterilized L-proline (Sigma, P5607) solution was added after autoclaving. The culture duration was 12 weeks.

Each treatment had three replicates and each replicate consisted of two Petri dishes, each with five explants per Petri dish, and each explant was made up of two halves of the plumule. The experimental design was an RCBD, and the experiment was blocked by time. The data on the percentage of callus and compact or friable EC formation were collected at the end of the experiment. The collected data were analysed using a two-way ANOVA.

### 4.8. Effect of Reduced Nutrients on the Formation of Friable and Compact EC (Experiment 5)

Isolated plumule halves were used to induce compact EC by culturing them in Petri dishes containing CIM for 12 weeks under dark conditions at 27 ± 1 °C. After 12 weeks, compact EC with an appearance as shown in Figure 1A was selected and used as explants, while non-EC was excised and discarded. These explants (*ca.* 50 mg) were then further cultured under the same culture conditions onto three different media (Table 6), including a Y3 medium, a GD medium [28], or a modified Y3 medium, which contained only 10% of the nitrogen (N), phosphorus (P), and potassium (K), and 10 mg L^−1^ thiamine hydrochloride (Table 7). All media were supplemented with 2,4-D (325 µM), AC (2.5 g L^−1^), sucrose (30 g L^−1^), and Gelzan (3 g L^−1^). The observation of callus, compact, or friable EC formation for each treatment was recorded after 8 weeks of the culture period. The explants (Sampoorna variety) were sub-cultured after 4 weeks onto the fresh medium of the same constituents, whereas no subculture was performed for IYD variety explants.

Each treatment consisted of three replicates and each replicate was a pair of Petri dishes with five explants per Petri dish. The experimental design was a completely randomized design (CRD) and the data collected were analysed using one-way ANOVA.

### 4.9. Histological Analysis

Compact EC and non-EC derived from plumules obtained from CAIRNS embryos were selected for a histological study using protocols recommended by the School of Biomedical Science Histology Facility, the University of Queensland. Prior to transport, the tissues were first fixed in a Karnovsky’s fixative [35] for 2 h, then washed under running distilled water for 20 min. The samples were then placed into cassettes, which were filled with 70% (*v*/*v*) ethanol and transported to the facility for processing. Upon arrival, the samples were dehydrated in a short series of ethanol solutions of increasing concentrations [70, 90, 100, and 100% (*v*/*v*)] and then covered with xylene twice, with each step applied for 45 min at room temperature. The samples were then covered in liquid paraffin wax (60 °C) for 45 min, then the treatment was repeated but the liquid paraffin wax-covered samples were placed into a vacuum chamber (0.4 MPa) for 45 min to ensure no air was trapped in the samples. Next, the cassettes containing the wax-treated samples were placed into a tray containing melted paraffin wax at the embedding station (Medite, TES Valida). The samples were removed from the cassettes in the paraffin tray and placed into the wax-filled metal containers, which were then put on the cooling plate. The tissues embedded in wax were removed from the metal containers after the wax was ice cold. The samples were kept in a freezer prior to sectioning into 6 µm sections using a rotary microtome (Leica Biosystems) and collected onto slides in a water bath (40 °C). The slides were left in the oven at 38 °C to remove moisture until performing the staining procedure.

In the staining procedure, the sections were first dewaxed in xylene for 2 min thrice and washed in a diluted series of ethanol concentrations of 100, 100, 90, and 70% (*v*/*v*), then, finally, running distilled water for 2 min for each step. The sections were now ready for Periodic Acid Schiff’s staining [36]. The sections were stained with 1% periodic acid (10 min), washed in distilled water (5 min), stained with Schiff’s reagent (20 min), and washed again in running distilled water (5 min). The slides were then covered with Mayer’s haemotoxylin (2 min), washed with distilled water (2 min), stained with 0.1% (*v*/*v*) ammonia (5 min), and finally rinsed in distilled water (2 min). The stained sections were then dehydrated in a series of ethanol solutions (70, 90, 100, 100, and 100%), submerged in each concentration for 2 min, and cleared in xylene thrice for 2 min in each step. The sections were then mounted with a coverslip using DePex mounting medium (Ajax) and left to dry overnight in a fume hood. The sections were observed under a light microscope with a camera attached (Olympus). The photos were taken at magnifications of 175×, 700×, and 780× using Olympus cellSens Imaging Software.

## 5. Conclusions

In summary, these studies have provided important information on the formation of compact EC, particularly the study on the influence of reducing the 2,4-D concentration (6 µM) for the transformation of compact EC into somatic embryos, which then formed plantlets. The benefit of using an amino acid mixture to produce compact EC is demonstrated, and the presence of starch granules in compact EC for coconut is first reported. Further studies need to be undertaken building on the learnings of the current study to produce friable EC, likely required for the establishment of a cell suspension culture system for coconut.

## Figures and Tables

**Figure 1 plants-12-00968-f001:**
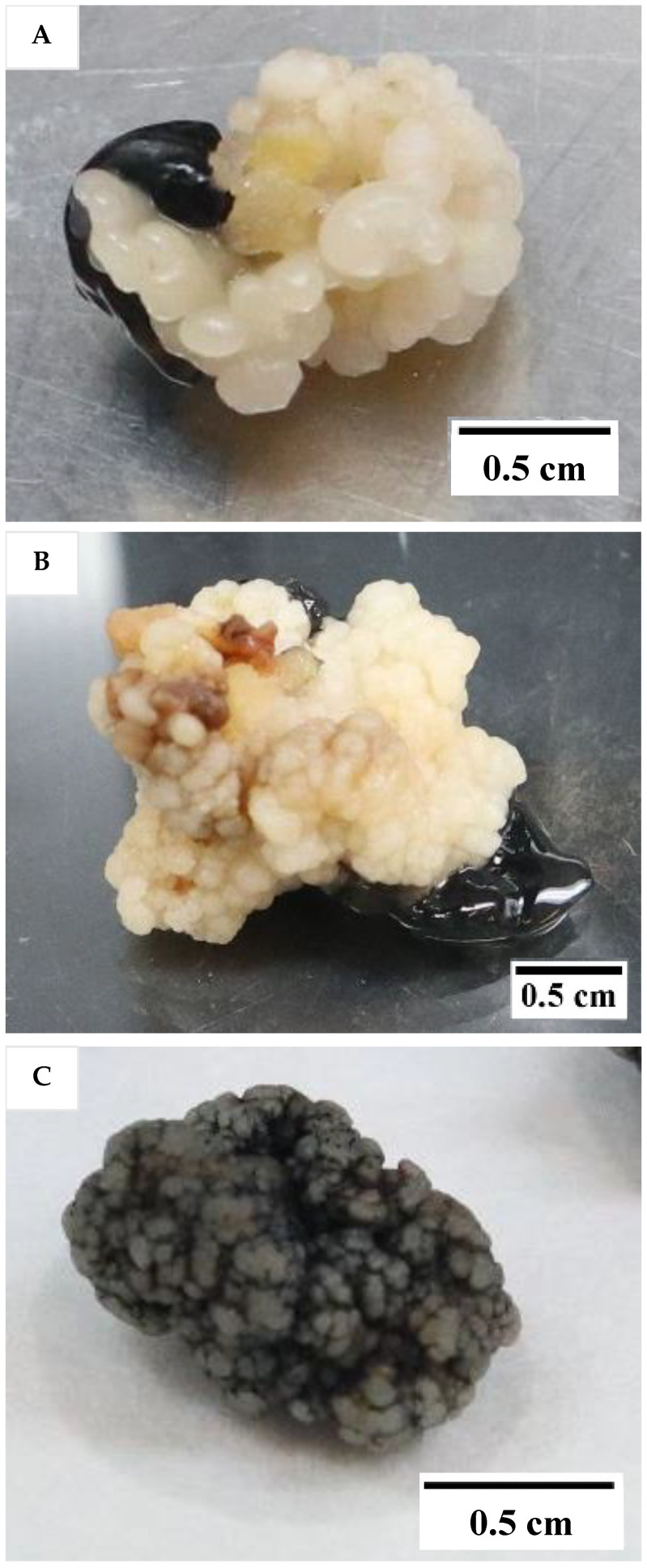
The type of compact callus formed. (**A**) Embryogenic callus (EC). (**B**) Non-EC. (**C**) Non-embryogenic callus appearing black in colour due to a coating of activated charcoal and produced from liquid medium.

**Figure 2 plants-12-00968-f002:**
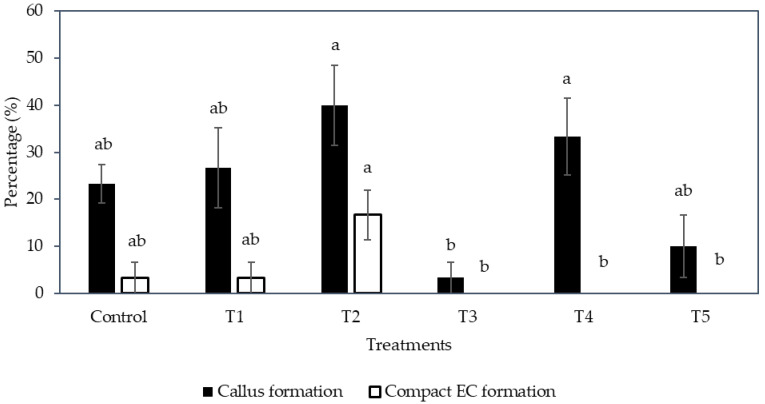
The effect of different medium types with various subculture frequencies (Control, T1 to T5, see Section 4.4) on the percentage of callus and compact embryogenic callus formed. The mean ± standard error for different treatments in each bar followed by different letters was significantly different for that parameter (Tukey’s test, *p* ≤ 0.05); *n* = 5.

**Figure 3 plants-12-00968-f003:**
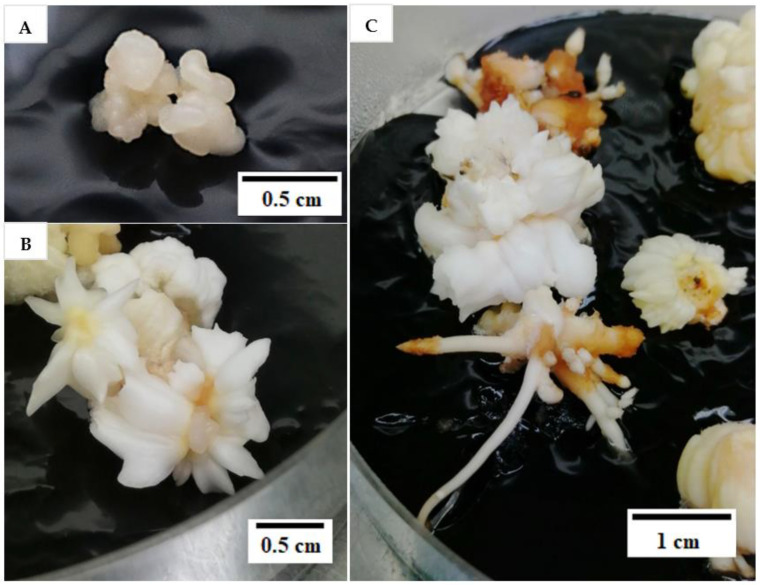
(**A**) Compact embryogenic callus (EC) used in the study to form friable EC. (**B**) Mature somatic embryos and (**C**) roots forming on compact EC after two subculture cycles.

**Figure 4 plants-12-00968-f004:**
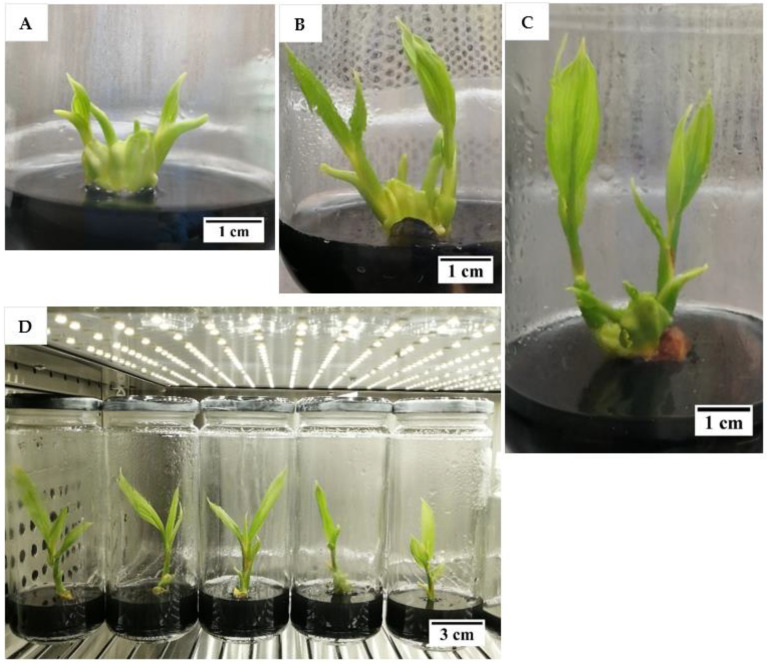
The development of Malayan Yellow Dwarf plantlets after (**A**) 4, (**B**) 5, (**C**) 7, and (**D**) 8 months on a plant growth regulator-free Y3 medium containing activated charcoal.

**Figure 5 plants-12-00968-f005:**
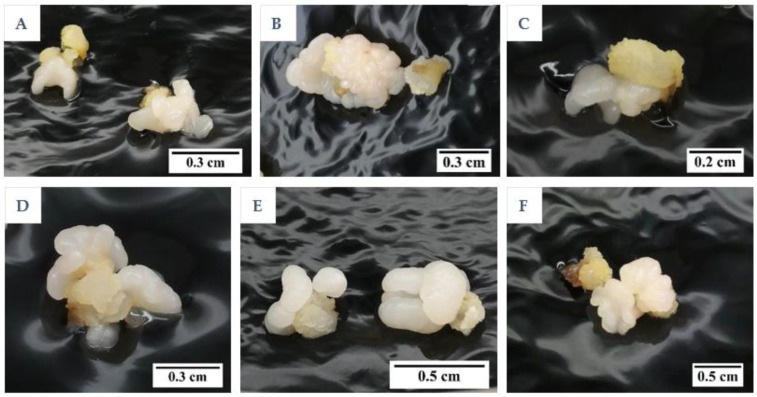
The formation of compact embryogenic callus for (**A**) control and treatments (**B**) T1, (**C**) T2, (**D**) T3, (**E**) T4, and (**F**) T5 (Control, T1 to T5; see table in Section 4.6) after 3 months.

**Figure 6 plants-12-00968-f006:**
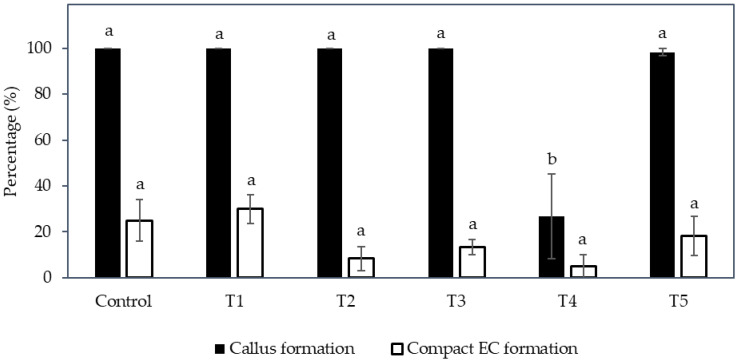
The effect of various combinations of nitrate and ammonium ion concentrations placed into the Y3 medium with and without amino acids (Control, T1 to T5; see table in Section 4.6) on the percentage of callus and compact embryogenic callus formation. Each bar represents the mean ± standard error; *n* = 5. Bars covered by the same letter were not significantly different (Tukey’s test, *p* ≤ 0.05).

**Figure 7 plants-12-00968-f007:**
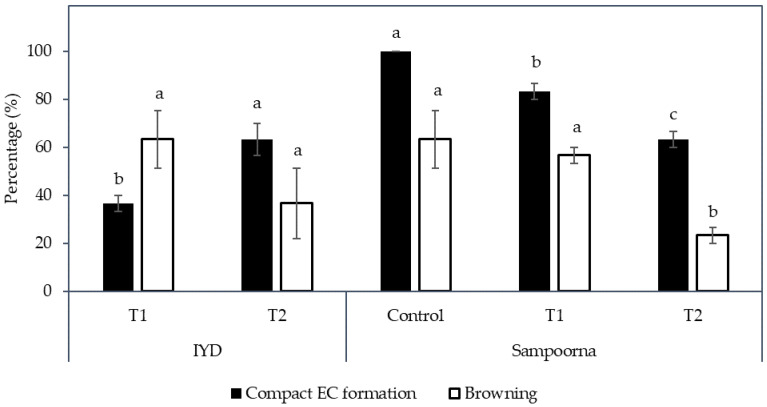
The effect of reduced nutrients (Control, T1, and T2) on the percentage of compact embryogenic callus (EC) formed and explants with necrosing (Browning) tissues. Mean value ± standard error is presented on each bar. The mean value of each treatment for each coconut variety followed by different letter was significantly different (Tukey’s test, *p* ≤ 0.05); *n* = 3. Control: Eeuwens Y3 medium; T1: Modified Eeuwens Y3 medium with only 10% N, P, and K and 10 mg L^−1^ thiamine hydrochloride; T2: Gresshoff and Doy (GD) medium.

**Figure 8 plants-12-00968-f008:**
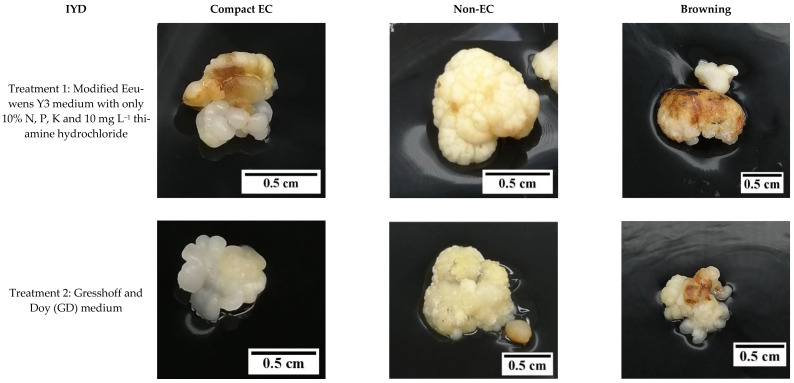
The effect of reduced nutrient concentration on the formation of compact embryogenic callus (EC), non-embryogenic callus (non-EC), and browning callus for the variety Indian Yellow Dwarf.

**Figure 9 plants-12-00968-f009:**
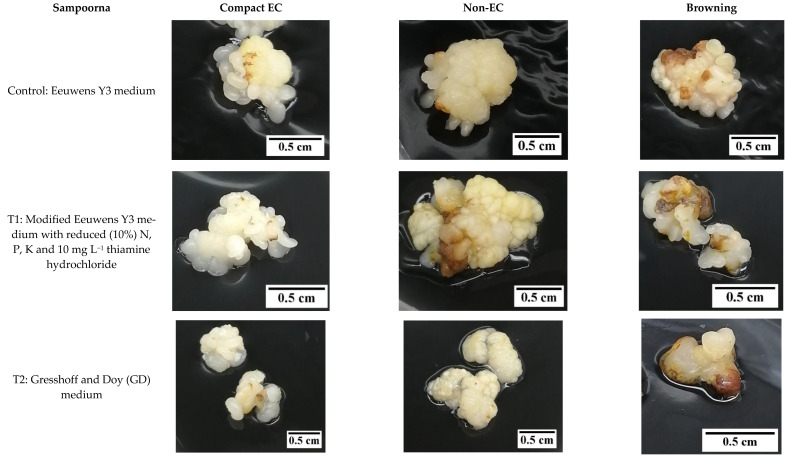
The effect of reduced nutrient concentration on the formation of compact embryogenic callus (EC), non-embryogenic callus (non-EC), and browning callus for the variety Sampoorna.

**Figure 10 plants-12-00968-f010:**
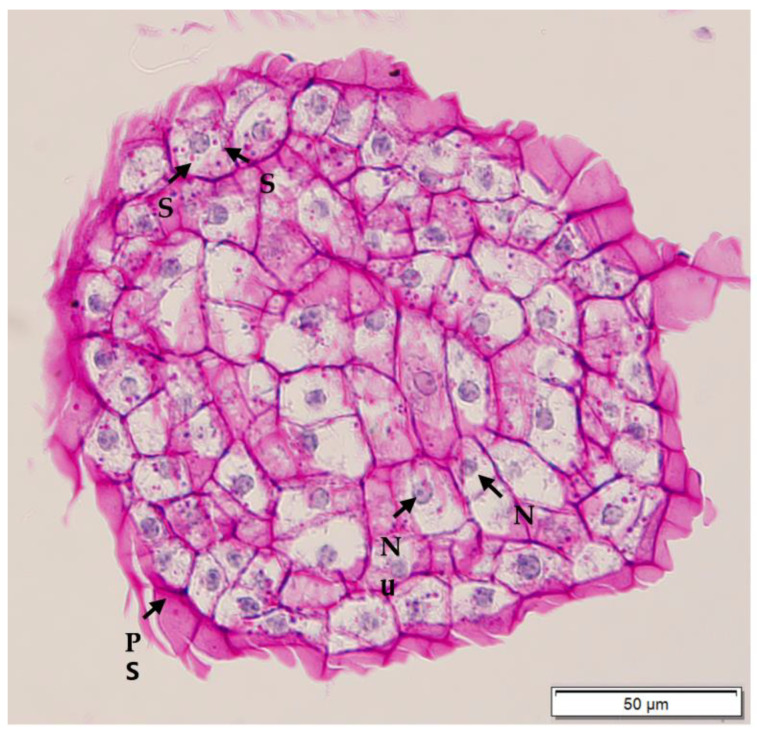
Histological section of compact embryogenic callus derived from a plumule obtained from a CAIRNS embryo, and after periodic acid Schiff’s staining. Nuclei were stained blue, while starch and polysaccharidic sheath (PS) were stained pink or purple. Numerous starch (S) granules are observed surrounding the large nucleus (N) with visible nucleolus (Nu).

**Figure 11 plants-12-00968-f011:**
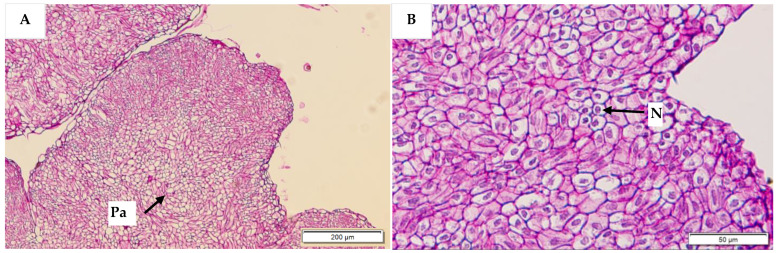
Histological sections of non-embryogenic callus stained with periodic acid Schiff’s reagent. Parenchyma (Pa) cells were observed in the central zone (**A**) at a magnification of 200 µm. No starch granules or meristematic cells were observed, under high resolution (**B**), but nuclei (N) were visible.

**Table 1 plants-12-00968-t001:** The effect of 2,4-dichlorophenoxyacetic acid (2,4-D) concentration on various tissue production after 28 weeks of culture.

Concentration of 2,4-D (µM)	Percentage of Explant (%) Producing				
	Necrotic Callus	Callus	Compact EC	Mature Somatic Embryos	Roots
600 (control)	20.0 ± 5.8 ^a^	80.0 ± 5.8 ^b^	32.5 ± 2.5 ^a^	0.0 ± 0.0 ^b^	0.0 ± 0.0 ^b^
6	2.5 ± 2.5 ^b^	97.5 ± 2.5 ^a^	0.0 ± 0.0 ^b^	57.5 ± 15.5 ^a^	20.0 ± 7.1 ^a^

Mean values ± standard error of different treatments in each column followed by different letters were significantly different (Tukey’s test, *p* ≤ 0.05); *n* = 4.

**Table 2 plants-12-00968-t002:** The effect of different treatments on the percentage of callus and compact embryogenic callus (EC) formed.

Treatment	Callus Formation (%)	Compact EC Formation (%)	Compact EC
Control: Eeuwens Y3 medium without L-proline	96.7 ± 3.3	13.3 ± 6.7	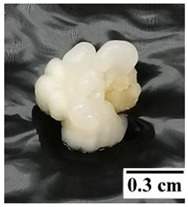
T1: Eeuwens Y3 medium + 6 mM L-proline	100.0 ± 0.0	0.0 ± 0.0	
T2: Chu (N6) medium + 6 mM L-proline	90.0 ± 5.8	0.0 ± 0.0	
T3: Modified Eeuwens Y3 medium + 6 mM L-proline	93.3 ± 6.7	6.7 ± 3.3	

The mean values (±standard error) of different treatments in each column were not significantly different (Tukey’s test, *p* ≤ 0.05); *n* = 3.

**Table 3 plants-12-00968-t003:** Treatments applied to the coconut plumules in Experiment 1, listing the medium type and subculture frequency used.

Treatment	Y3 Medium Type	Subculture Frequency
Control	Semi-solid	Step 1: No subculture for 12 weeks Step 2: No further subculture for 20 weeks
T1		Subculture every 4 weeks for 20 weeks
T2		No subculture for 20 weeks
T3	Liquid	
T4		Subculture every 2 weeks for 20 weeks
T5		Subculture every 4 weeks for 20 weeks

**Table 4 plants-12-00968-t004:** The combinations of different nitrate and ammonium ion concentrations that were added to the Y3 medium, and with and without an amino acid mixture and applied to plumule tissues.

Treatment	Amino Acid Mixture	Concentration of KNO_3_ (Mm)	Concentration of NH_4_Cl (Mm)	Ratio of NO_3_^−^/NH_4_^+^
Control	No	20	10	2:1
T1	Yes			
T2	No	40		4:1
T3	Yes			
T4	No		20	2:1
T5	Yes			

**Table 5 plants-12-00968-t005:** Different treatments for the induction of friable embryogenic callus (EC) from plumule tissues cultured in either Y3 or N6 medium.

Treatment	Friable EC Induction Medium
Control	Y3 medium without L-proline
T1	Y3 + 6 mM L-proline
T2	N6 + 6 mM L-proline
T3	Modified Y3 (ammonium sulphate replacing ammonium chloride) + 6 mM L-proline

**Table 6 plants-12-00968-t006:** Testing of different treatments using Eeuwens Y3 medium, modified Eeuwens Y3 medium, or Gresshoff and Doy (GD) medium to induce friable EC.

Treatment	Medium
Control	Y3 medium (control)
T1	Modified Y3 medium with only 10% N, P, and K and 10 mg L^−1^ thiamine hydrochloride
T2	GD medium

**Table 7 plants-12-00968-t007:** The differences in the make-up of the standard and modified Y3 formulations.

Chemicals	Y3 Medium (mg L^−1^)	Modified Y3 Medium with Only 10% N, P, K (mg L^−1^)
Potassium nitrate (KNO_3_)	2020	202.0
Potassium chloride (KCl)	1492	149.2
Ammonium chloride (NH_4_Cl)	535	53.5
Sodium dihydrogen ortho-phosphate (NaH_2_PO_4_·2H_2_O)	312	31.2
Thiamine hydrochloride	1	10.0

## Data Availability

All data are included in the main text.

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
