# Peer review of "Coconut Callus Initiation for Cell Suspension Culture"

_plants, 2023, doi:10.3390/plants12040968_

Round 1
Reviewer 1 Report
It is a conscientious methodological paper which may be published. This investigation doesn't have positive results because the main problem - the obtaining of friable callus- have not been solved. However, negative results should also be published in order to know which ways are wrong
Reviewer 2 Report
The manuscript (Plants-2201276) entitled “Coconut callus initiation for cell suspension culture” demonstrates an efficient protocol to induce callus by using cell suspension cultures derived from mature embryos.
· The authors selected different varieties from Australia, India, the Philippines, and Vietnam.
· The present study also demonstrates the successful application of various media such as Y3, N6, and GD and the addition of sucrose, activated charcoal, and gelzan (a substitute for agar).
· This study also demonstrates that the addition of L-proline plays a significant role in the induction of embryogenic suspension cultures in popular coconut varieties.
· The manuscript is an excellent contribution to developing embryogenic suspension cultures in commercial varieties of coconut and their mass propagation.
· Keywords should be in alphabetical order.
· The English language of the manuscript needs to improve.
· Whether the authors perform any genetic fidelity studies in regenerated plants to check the somaclonal variation, if performed, please include the data in the revision.
The submitted manuscript may be acceptable for publication after a minor revision.
